# Olfactory and Gustatory Outcomes Including Health-Related Quality of Life 3–6 and 12 Months after Severe-to-Critical COVID-19: A SECURe Prospective Cohort Study

**DOI:** 10.3390/jcm11206025

**Published:** 2022-10-12

**Authors:** Elisabeth Arndal, Anne-Mette Lebech, Daria Podlekarava, Jann Mortensen, Jan Christensen, Frederikke F. Rönsholt, Thomas Kromann Lund, Terese L. Katzenstein, Christian von Buchwald

**Affiliations:** 1Department of Otorhinolaryngology, Head and Neck Surgery and Audiology, Rigshospitalet, Copenhagen University Hospital, 2100 Copenhagen, Denmark; 2Department of Infectious Diseases, Rigshospitalet, Copenhagen University Hospital, 2100 Copenhagen, Denmark; 3Department of Clinical Physiology and Nuclear Medicine, Rigshospitalet, Copenhagen University Hospital, 2100 Copenhagen, Denmark; 4Department of Occupational- and Physiotherapy, Rigshospitalet, Copenhagen University Hospital, 2100 Copenhagen, Denmark; 5Department of Cardiology, Rigshospitalet, Section for Lung Transplantation, Copenhagen University Hospital, 2100 Copenhagen, Denmark

**Keywords:** olfaction, gustation, COVID-19, parosmia, dysgeusia, HRQoL, TDI, BSIT, SIT

## Abstract

**Background:** Long-term follow-up studies of COVID-19 olfactory and gustatory disorders (OGDs) are scarce. OGD, parosmia, and dysgeusia affect health-related quality of life (HRQoL) and the ability to detect potential hazards. **Methods:** In this study, 29 patients reporting OGD 1 month after severe-to-critical COVID-19 were tested at 3–6 months and retested at 12 months in case of hyposmia/anosmia. We used Sniffin Sticks Threshold, Discrimination, and Identification (TDI) test, Sniffin Sticks Identification Test (SIT16), Brief Smell Identification Test (BSIT), taste strips, and HRQoL. The patients were part of the prospective SECURe cohort. **Results:** Overall, 28% OD (TDI), 12% GD, 24% parosmia, and 24% dysgeusia (questionnaire) at 3–6 months (*n* = 29) and 28% OD (TDI), 38% parosmia, and 25% dysgeusia (questionnaire) at 12 months (*n* = 8) were observed. OGD decreased HRQoL: For 13%, it had a negative effect on daily life and, for 17%, it affected nutrition, 17% reported decreased mood, and 87–90% felt unable to navigate everyday life using their sense of smell and taste. A comparison of SIT16 and BSIT to TDI found sensitivity/specificity values of 75%/100% and 88%/86%. **Conclusion****s****:** This is the first study to examine TDI, SIT16, BSIT, taste strips, and HRQoL up to 1 year after severe-to-critical COVID-19. The patients suffering from prolonged OGD, parosmia, and dysgeusia experienced severely decreasing HRQoL. We recommend including ear–nose–throat specialists in multidisciplinary post-COVID clinics.

## 1. Introduction

Previous studies have reported that 19–98% of COVID-19 patients complained of olfactory and gustatory dysfunction (OGD) during the initial phase of the infection, and as many as 32% suffered from parosmia [1,2,3,4]. For comparison, approximately 15% of the public in the western world have hyposmia and 5% anosmia [5,6], so the OGD prevalence is soaring upwards due to COVID-19. OGD may be caused by multiple aetiologies such as sinonasal disease, global airway diseases such as asthma with nasal polyps and chronic obstructive pulmonary disease, trauma, infection, age, neurodegenerative diseases, or other medical, iatrogenic, congenital, toxic, and idiopathic causes, but here, we focus on COVID-19. OGD affects women more often than men and can be the only symptom of COVID-19. However, it is often seen in combination with COVID-19 symptoms such as cough, fever, and muscle soreness [7]. OGD has been reported in patients with both mild, moderate, and severe COVID-19 [8,9]. OGD is up to 28 times as common in patients with COVID-19, compared with healthy controls, and three times as common compared with patients with influenza [2,10]. The olfactory mucosa is more vulnerable to SARS-CoV-2 than the adjacent respiratory mucosa possibly due to a higher virus load when air filters through the nose during the initial stages of infection, but the exact mechanism behind OGD is not fully elucidated [11]. COVID-19 affects the olfactory sustentacular cells [12], ACE2 receptors in the olfactory mucosa, and taste buds, and autopsy findings have shown focal leucocyte infiltration and atrophy, suggesting axonal damage [13,14,15]. COVID-19 may also be neurogenic, capable of the cerebral affection of the olfactory pathways such as the olfactory bulb and the orbitofrontal cortex [16,17]. As the COVID-19 pandemic continues, our knowledge of post-COVID syndromes (COVID-19 symptoms present for more than 12 weeks after the initial infection) has increased [18]. Recently studies have reported ODG in 1.6–27% up to 5 months after COVID-19. However, long-term follow-up data based on validated olfactory and gustatory testing are sparse [19,20,21].

OGD can severely affect patients, especially if they have prolonged symptoms such as parosmia and anosmia. Parosmia describes the distortion of normal olfactory stimuli so that they are registered, most often, as vile. OGD can impact all aspects of life, from health-related quality of life (HRQoL) [22], nutritional state [23,24], and social interaction to mother–child bonding [25] and the choice of partner [26]. OGD may impact livelihood if a patient’s job is dependent upon their sense of smell and taste and may expose patients to a potentially hazardous situation in the case of fire, the consumption of spoiled foods, or exposure to chemical fumes [27]. We have seen a massive increase in the number of patients referred due to post-COVID OGD requiring diagnosis, treatment, and follow-up at specialised otorhinolaryngologic outpatient clinics.

The aims of the current study were (1) an evaluation of long-term OGD and HRQoL follow-up of previously hospitalised COVID-19 patients using validated psychophysical olfactory and gustatory tests and (2) a comparison of the Sniffin Sticks Identification Test (SIT16) and Brief Smell Identification Test (BSIT), two olfactory identification tests, with the gold standard Sniffin Sticks Threshold, Discrimination, and Identification (TDI) test.

## 2. Materials and Methods

We included 29 consecutive patients from March to May 2020 at Rigshospitalet, Copenhagen University Hospital, Denmark. The patients were a subgroup of the prospective SECURe cohort (Sequelae COVID-19, University Hospital, Rigshospitalet) [28]. The SECURe cohort included those patients with severe-to-critical COVID-19 in need of hospitalisation and, in some cases, admittance to the intensive care unit (ICU) and mechanical ventilation. All patients had a positive PCR SARS-CoV-2 test at admittance; the strain was not recorded, but the Delta variant was the dominant variant at inclusion time. All the patients in the SECURe cohort were asked via telephone about their subjective loss of smell and taste 1 month after discharge. The patients answering yes to a decreased sense of smell and taste were included in the present study. All the patients were examined in a single visit 3–6 months after discharge. The examination included questions on HRQoL and a visual analogue scale (VAS; 0, no distortion; 10, worst possible distortion) about their subjective sense of smell and taste prior to psychophysical olfactory and gustatory testing. The HRQoL questions included the retrospective question concerning the “subjective sense of smell and taste before and during COVID-19”. The patients were also asked about HRQoL at the time of the 3–6 months visit: their subjective loss of smell and taste and its effect on their daily life, nutrition, mood, ability to navigate everyday life using their sense of smell and taste, and subjective ability to smell and taste certain odours (sweat, perfume, smoke) and tastes (coffee, bitter, sweet, sour, salt). The patients were asked about the presence of parosmia and dysgeusia. Psychophysical olfactory testing was performed using the validated Burghart Sniffin Sticks Threshold (T), Discrimination (D), and Identification (I) test with the validated Danish multiple-choice answers for the Identification (I) subtest and the Danish Version of the Brief Smell Identification Test (BSIT) [29,30,31]. The Identification test is named SIT16 when it is used separately for testing olfactory identification ability. Both nostrils were tested simultaneously, and we used the “reading first” test condition based on which the multiple-choice answers were read before the odour was introduced [32]. The psychophysical gustatory screening was performed using a brief version of Burghart’s taste strips (the highest concentration of sweet, sour, bitter, and salty) [33]. All the tests were carried out from 8 am to 3 pm in a well-ventilated and odourless room and according to manufacturers’ instructions. The odours in TDI, SIT16, and BSIT are predefined by the manufacturer, making them reliable and reproducible tests. The patients were asked to abstain from eating and drinking 30 min prior to testing. An otorhinolaryngologist (EA) performed a flexible nasal endoscopy of the nasal cavity and olfactory region after testing. Olfactory function scores were as follows: normosmia: TDI ≥ 30.75, SIT16 > 11, BSIT > 9; hyposmia: TDI 17–30.74, SIT16 9–11, BSIT 8-6; anosmia: TDI ≤ 16, SIT16 ≤ 8, BSIT ≤ 5 [30]. Gustatory function scores were as follows: normogeusia: 4 out of 4 correctly identified tastes, and hypogeusia: 0–3 out of 4 correctly identified tastes. Patients’ olfactory function was categorised according to their TDI results. The BSIT and SIT16 results were used for comparison with the gold standard TDI test. Any patients with hyposmia and anosmia according to the psychophysical test results (TDI) were seen at a 12-month follow-up visit and offered high-intensity olfactory training with four essential oils (of the patients’ own choice) containing everyday odours twice daily for 3 months [34,35]. According to the general practice at our department, patients with hyposmia/anosmia were offered an MRI scan of the cerebrum or a CT scan if there were contraindications to an MRI. At the 12-month follow-up visit, the patients had a full TDI test and were asked about HRQoL (the presence of parosmia and dysgeusia). Written informed consent was collected from all patients. The study was approved by the local ethic committee (H-20028792) and complies with the Declaration of Helsinki for Medical Research involving Human Subjects.

### 2.1. The SECURe Cohort

This prospective SECURe study aimed at examining the long-term sequelae of COVID-19. The study was conducted by a multidisciplinary research group including ear, nose, and throat, infectious medicine, cardiology, occupational and physical therapy, and clinical physiology. The variables relevant to each medical specialty were included and planned for individual publication. In this article, we included ear, nose, and throat variables (mentioned above) and variables relevant to the description of a cohort after severe-to-critical COVID-19 (age*, gender*, disease severity*, CCS, treatment during hospitalisation*, ICU admission and duration of hospitalisation, lung function*, gastroesophageal reflux*, smoking status*, concentration difficulties, and headache), which are listed in Table 1. The variables marked by a * were chosen, as they may influence the olfactory function by affecting the nasal airflow and/or olfactory mucosal health. Concentration difficulties and headaches were included, as they may affect the patient’s ability to perform psychophysical olfactory testing. Treatments with remdesivir and dexamethasone were recorded, as they may affect the regeneration of the olfactory mucosa. Demographic data were extracted from patients’ electronic medical records, while lung function testing was performed at the post-discharge visit. All the tests were carried out at Rigshospitalet, Copenhagen University Hospital, Denmark.

### 2.2. Statistical Analysis

Continuous variables are listed as mean (SD) and categorical and binary variables as proportion. An independent *t*-test was used to compare means, while Pearson’s chi-squared and Fisher’s exact tests were used to compare percentages. A paired sample *t*-test was used to compare the mean TDI test scores at 3–6 and 12 months for patients with hyposmia/anosmia. A *p*-value < 0.05 was considered statistically significant. We used IBM Corp. Release 2017 IBM SPSS Statistics for Windows, Version 25.0 (Armonk, NY, USA: IBM Corp.).

## 3. Results

The mean age was 56 years (SD 12.7), and the patients were predominantly male (62%), with a mean duration of hospitalisation of 17 days (SD 15.0). All the study participants were tested 3–6 months (median 4.7 months, SD 0.7) after discharge, and those with hyposmia/anosmia were retested at 12 months after discharge (Table 1).

Except for the variable “concentration difficulties”, there was no statistically significant difference between demographic and comorbidity variables in the normosmic and hyposmic/anosmic groups. There was a trend that patients in the OGD group had been admitted more to the ICU than the normosmic group (Table 1).

### 3.1. HRQoL: Subjective Sense of Smell before and during COVID-19 (n = 29)

Before COVID-19, 3% reported a very good sense of smell, 90% reported a normal/good sense of smell, and 7% reported decreased/bad sense of smell. None of the patients reporting OD before COVID-19 had OD on the TDI test. During COVID-19, 0% reported a very good sense of smell, 3% reported a normal/good sense of smell, and 97% reported decreased/bad sense of smell.

#### HRQoL: Subjective Sense of Taste before and during COVID-19 (*n* = 29)

Before COVID-19, 3% reported a very good sense of taste, 94% reported a normal/good sense of taste, and 3% reported decreased/bad sense of taste. The patients reporting GD before COVID-19 did not have GD on taste strip testing. During COVID-19, 0% reported a very good sense of taste, 3% reported a normal/good sense of taste, and 97% reported decreased/bad sense of taste.

### 3.2. HRQoL at 3–6 Months after Hospitalisation for COVID-19 (n = 29)

Overall, 87–90% of the patients reported that their decreased sense of smell and taste after their COVID-19 infection negatively affected their ability to navigate everyday life (e.g., smell their own body odour, the freshness of food, and smell smoke) (black box, Figure 1). Of all the patients, 13% reported that their OGD negatively affected their daily life, 17% reported affected nutrition, and 17% pointed to the deterioration of their mood.

### 3.3. HRQoL: Distorted Sense of Smell and Taste at 3–6 Months after Hospitalisation for COVID-19 (n = 29)

Parosmia was reported by 24% of all the patients, with a mean VAS score of 6.9 (SD 2.7) (VAS, 0 = no distortion; 10 = worst possible distortion). Parosmia was reported by 24% (*n* = 5) in the normosmic group and 25% (*n* = 2) in the hyposmic/anosmic group. When asked to respond to the statement “I can’t smell the following odour”, the patients replied: sweat 28%, coffee 17%, and perfume 14%. Dysgeusia was reported by 24% of all the patients, with a mean VAS score of 7.0 (SD 2.5). Dysgeusia was reported by 24% (*n* = 5) in the normosmic group and 25% (*n* = 2) in the hyposmic/anosmic group. When asked to respond to the statement “I can’t taste the following taste”, patients replied: coffee 17%, bitter 10%, salt 3%, sweet 7%, and sour 7%.

### 3.4. Psychophysical Olfactory and Gustatory Testing 3–6 Months after COVID-19 (n = 29)

Overall, 72% of the patients were normosmic, while 28% had OD. The Threshold (T) scores were affected more than the Discrimination (D) and Identification (I) scores. The gustatory function was decreased in 12% of the patients (Table 2, Figure 2).

#### 3.4.1. Imaging

Six out of eight patients with olfactory dysfunction (hyposmia and/or anosmia) had an MRI of the cerebrum. All scans were normal (olfactory region and olfactory cortex). The remaining two patients declined both MRI and CT.

#### 3.4.2. Sinonasal Disease

Two patients in the normosmic group suffered from chronic rhinosinusitis (CRS), while one patient in the OD group suffered from hyposmia and CRS and a large unilateral septal deviation. This patient reported a normal sense of smell before COVID-19 and had a post-COVID-19 TDI score of 27 (hyposmia), BSIT score of 7, and parosmia and dysgeusia. None of the anosmic patients had CRS.

#### 3.4.3. Psychophysical Olfactory Testing 12 Months after Hospitalisation for COVID-19

All the patients with OGD at 3–6 months (*n* = 8) had persistent, decreased olfaction at the 12-month visit (Table 3). Again, the T scores were lower than the D and I scores. Parosmia was reported by 38% of the patients and dysgeusia by 25%.

#### 3.4.4. Olfactory Training

Six out of eight patients (75%) with OD performed olfactory training during the follow-up period. Patient 5 (P5) with hyposmia and P7 with anosmia had not performed olfactory training. We found a mean TDI score of 21.1 at 3–6 months and 21.6 at 12 months (*p* = 0.87). We found the following differences in TDI scores between 3–6 months and 12 months for the eight patients with olfactory dysfunction: ∆P1: −11.5, ∆P2: +19.2, ∆P3: −5.5, ∆P4: +2.0, ∆P5: −3.0, ∆P6: −3.8, ∆P7: +2.0, and ∆P8: +3.2. Three patients (+ training) had a change in TDI score which exceeded the minimal important difference of 5.5 [36], and only one patient had an improvement in olfaction.

### 3.5. Frequency of Incorrect Answer per Odour in Identification Test

The mean (SD) incorrect answer per odour in SIT16 was 18.5% (14.8) with one outlier: cinnamon (70% incorrect answers). The mean (SD) incorrect answer per odour in BSIT was 24.6% (10.2) with two outliers: cherry (47% incorrect answers) and lemon (40% incorrect answers).

### 3.6. Sensitivity and Specificity of the Individual Psychophysical Identification Tests (BSIT and SIT16) Compared with the Gold Standard TDI Test

We found a negative predictive value of 94.7% for the BSIT and 91% for the SIT16 test (Table 4a,b).

## 4. Discussion

Our study combined TDI, SIT16, BSIT, taste strip testing, HRQoL, and clinical examination of 29 patients after hospitalisation for severe-to-critical COVID-19. We found that 28% (8/29) of the patients still suffered from OGD, parosmia, dysgeusia, and reduced HRQoL at 3–6 months and observed no improvement in olfactory function and parosmia after 12 months. Some studies of hospitalised patients with acute moderate-to-severe COVID-19 have shown 58–62% OGD [21,37]. Previous studies of the olfactory and gustatory function at 3–6 months post-COVID-19 in mixed patient populations (hospitalised and non-hospitalised) using different olfactory and gustatory psychophysical tests have shown a broad range of 27–60% OD and 10% GD [38,39,40,41]. Our findings of OGD at 3–6 months after hospitalisation for severe-to-critical COVID-19 were in the lower range of what has previously been reported for non-hospitalised patients, as mentioned above. This was in accordance with previous findings according to which OGD prevalence was inversely correlated to COVID-19 infection severity [42]. Our findings of 28% OD at 12 months were in accordance with the only previous 12-month follow-up study by Vaira et al. [19]. Our data support the sparse evidence, indicating that the regenerative potential and spontaneous recovery rate are decreased in those patients suffering from post-COVID-19 OGD.

Parosmia was reported by 24–38% of the patients and dysgeusia by 24–25% at 3–6 and 12 months with psychophysical testing, showing 28% OD and 12% GD. Other studies have reported 43–75% parosmia at 1–6 months [43,44]. Parosmia may be a sign of returning function, which has not yet been fully achieved, or it may be a sign of continued olfactory damage [44,45]. Psychophysical testing for parosmia has become available but is not widely used, and the fMRI of the olfactory cortex is also not routinely used but may hold a key to understanding parosmia [46,47]. We found some, albeit not consistent, evidence of COVID-19 affecting some specific odours (cinnamon, cherry) more than others. Boscolo-Rizzo et al. found significant differences between the discrimination rate of all but 3 of the 34 UPSIT odours when comparing anosmic with normosmic patients, with the most affected odours being cinnamon, peach, watermelon, menthol, and coconut [38]. These inconsistent findings may be explained by the reports that SARS-CoV2 only targets the supporting cells and not the actual olfactory receptor itself which may cause a random affection of the olfactory epithelium.

We found that 12% had GD at 3–6 months, affecting only bitter and salty receptors (7% and 3%, respectively; Table 2). The prevalence was higher than the previous findings of GD with 0–10%, but the affection of specific tastes has not been reported previously [38,40]. Bitter tastants activate the same taste receptor (Type II) as sweet and umami, while salty tastants affect the same taste receptor as sour (Type III), so our finding did not seem to be receptor-specific [48]. Bitter receptors are reported to recognise pathogenic bitter products and activate the innate immune system in the respiratory tract [49]. The exact mechanisms behind OGD, parosmia, and dysgeusia are not fully known. Our study supports the finding that patients with early OGD still suffer from parosmia even one year after their COVID-19 infection, underlining the need for additional research within this field.

We observed a marked negative effect on HRQoL in those patients suffering from OGD, parosmia, and dysgeusia at 3–6 and 12 months (Figure 1). Parker et al. found a significant correlation between the strength of a parosmic odour and the level of disgust [23]. The more disgusting the odour, the larger the impact on HRQoL [23]. The key food triggers of parosmia included common everyday items such as coffee, animal products, onion, and chocolate [23]. The review by Saniasiaya et al. found that patients with post-COVID-19 OGD expressed 57% safety-related concerns, and 84.6% indicated diminished enjoyment of food, with 66.5% highlighting the loss of appetite [50]. Vaira et al. found a significant reduction in the mental components of the general health questionnaire SF-12 in long COVID patients with OGD [51]. The prevalence of long-term (>1 year) symptoms reducing HRQoL amongst these patients remains to be explored, and patients should be offered guidance on appropriate coping mechanisms [50].

We also observed the previously described discrepancy between a subjective sense of smell and taste and psychophysical testing, with 87–90% reporting decreased HRQoL due to their subjective OGD, compared with 28% having OD according to the test scores (Figure 1 and Figure 2) [6,20]. We suggest that the subjective decreased sense of smell despite a normal olfactory test may be caused by a primary decrease in the odour threshold where patients are unable to smell subtle odours while still having normal odour discrimination and identification, which are both suprathreshold subtests [29,30]. Additionally, there may be individual differences in the cerebral interpretation of the olfactory signal [47]. The subjective decreased sense of taste is often caused by decreased olfaction, but the misconception may arise from the fact that laymen attribute the perception of taste not only to the basic tastes (sweet, salty, sour, bitter, and umami) but also to olfactory stimuli. This underlines the importance of using validated psychophysical olfactory and gustatory testing when examining patients.

Due to the small sample size and the relatively late onset of training, we were not able to reach a conclusion regarding the possible effects of olfactory training (OT) initiated at 3–6 months post-COVID-19. A review and meta-analysis of very heterogenous OT regimes of different durations in post-viral OD (before COVID-19) showed a significant effect of training [36]. However, training efficacy may be linked to aetiology and patient subgroups [52]. Presently, only one study of early OT, on average 5 weeks after the primary COVID-19 infection in OD patients, has been published [53]. It compared OT to OT and oral corticosteroids (OCS) in 27 post-COVID-19 patients. Only the nine patients in the OT+OCS group had a significant improvement in TDI scores after a median of 10 weeks. Our results support previous findings indicating that COVID-19 may cause more severe or longer-lasting damage to the olfactory mucosa than other viruses but may also be dependent upon the SARS-CoV-2 strain [1,54].

In comparison with previous findings, our OGD patients were not statistically different compared with the normosmic group [7]. This may have been affected by the sample size in the hyposmic/anosmic group. The trend towards a higher percentage of intensive care unit (ICU) admittances was seen in the OGD group. Sayin et al. reported that 42.3% of COVID-19 patients reported a subjective sense of smell and taste loss upon discharge from the ICU (mean stay of 10 days) [55]. The patients with hyposmia/anosmia suffered significantly more from concentration difficulties (50%) than normosmic patients (19%). The level of parosmia and dysgeusia was similar in the two groups, so the difference in concentration capability did not seem to influence this quality of smell and taste. Hassett et al. reported up to 59% cognitive dysfunction at 2 months after discharge, which was correlated to COVID-19 severity [56]. However, it is not known whether the concentration difficulties decrease the patient’s ability to complete the psychophysical test, resulting in lower scores, or if OGD and cognitive difficulties are both parts of post-COVID-19syndromes [16,17,57]. The cognitive dysfunction seen in our patients may solely be caused by COVID-19, but there may also be an effect of the post-intensive care syndrome seen in ICU-admitted patients [58,59].

There was no difference in the percentage of active and former smokers and no difference in FEV1% predicted (forced expiratory volume in 1 s) between the normosmic and hyposmic/anosmic groups (Table 1). Therefore, we found no sign of OD being caused by an altered respiratory (nasal versus oral respiration) pattern [60]. Due to the timing of our study, none of the included patients received dexamethasone during hospitalisation, and only one patient in each group (normosmic versus hyposmic/anosmic) received the anti-viral drug remdesivir. The olfactory function can also be affected by simultaneous CRS. As mentioned in the Results section, one patient in the OD group suffered from hyposmia + CRS + a large unilateral septal deviation. This could have affected the olfactory function, but the patient reported a normal sense of smell before COVID-19, thus suggesting that the hyposmia was most likely caused by COVID-19. If this patient had been excluded, OD prevalence would have decreased from 28% to 24%. We chose to keep this patient in the analysis. Besides the abovementioned patients, a normal nasal endoscopy was seen in 63% of the OD patients compared with 53% in the normosmic group. Therefore, we did not find the endonasal anatomy to affect the olfactory results.

We compared both olfactory screening tests (SIT16 and BSIT) with the full TDI test and found both the SIT16 and BSIT to be valid screening tools with high sensitivity and specificity (Table 4). When choosing a screening test, it is important that the test correctly identifies those with a normal olfactory function given a normal test result so that patients with an abnormal result can be referred for more accurate TDI testing. A normal BSIT result is truly normal in 95% of cases, and a normal SIT16 result is truly normal in 91% (negative predictive value) of cases, while an abnormal BSIT result is truly abnormal in 70% of cases, and an abnormal SIT16 result is truly abnormal in 100% of cases (positive predictive value). Taken together, this means that the BSIT overestimates OD, and SIT16 slightly underestimates OD. Thus, the BSIT will result in more patients with normosmia being TDI-tested than the SIT16.

There are pros and cons for each test, but the most important factor for clinicians is to ask, test, and refer for full psychophysical testing and ear, nose, and throat evaluation.

Our study has some strengths and limitations. This is the first study to examine TDI, SIT16, BSIT, Taste Strips, HRQoL, and the effects of olfactory training up to one year after severe-to-critical COVID-19 infection. All patients underwent both ear–nose–throat and pulmonological evaluations, including flexible endoscopic nasal examination and lung function testing. Extensive demographic data are available on all patients, as they are part of a prospective cohort study (the SECURe study). Our study is limited by the relatively small number of patients, and as mentioned, care should be taken when interpreting the results, including those of OT. We had a male prevalence in the normosmic group which may represent a selection bias, but the gender distribution was even in the hyposmic/anosmic group. It would also have been interesting to have had baseline psychophysical test scores for comparison with the long-term test scores and follow the rate of spontaneous regeneration over time. Baseline testing instead of the one-month post-discharge phone interview could have better identified patients with OGD. The gustatory function was tested using the brief version of the taste strips, which contains the highest concentration of the four basic tastants sweet, sour, salty, and bitter. The use of the extended taste strips using four decreasing concentrations of each tastant may have yielded additional information.

## 5. Conclusions

This study, for the first time, examined several measures of olfaction and gustation, including TDI, SIT16, BSIT, taste strips, and HRQoL, as well as the effect of OT up to 1 year after severe-to-critical COVID-19. We found that up to 28% of the patients suffered from prolonged OGD, and up to 38% experienced parosmia and dysgeusia with severe negative effects on HRQoL. We support the use of validated olfactory and gustatory testing, as our results revealed that both the BSIT and SIT16 olfactory screening tests have high sensitivity and specificity. A persistent high number of patients with OGD due to any cause may be expected, also considering the reports of COVID-19 reinfection even after complete vaccination and new COVID variants with different OGD potentials [54,61,62]. We recommend including ear–nose–throat specialists in multidisciplinary post-COVID outpatient clinics.

## Figures and Tables

**Figure 1 jcm-11-06025-f001:**
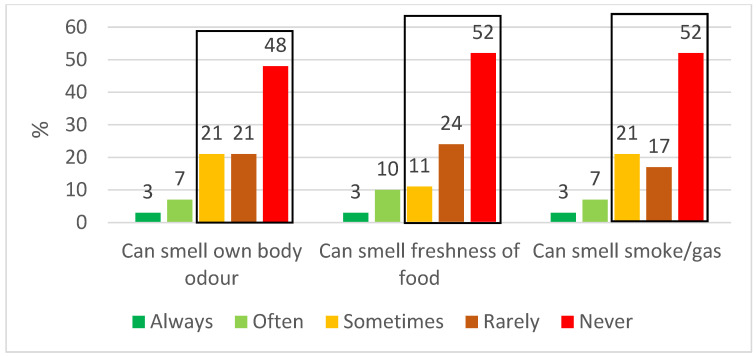
HRQoL; patients’ ability to navigate everyday life using their sense of smell and taste 3–6 months after hospitalisation for COVID-19 (*n* = 29). The black box shows the percentage of patients where this ability was decreased.

**Figure 2 jcm-11-06025-f002:**
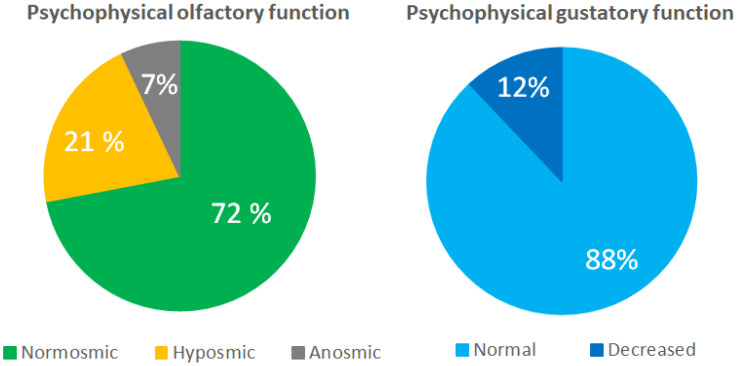
Psychophysical olfactory (Threshold, Discrimination, and Identification) and gustatory function (taste strips) 3–6 months after hospitalisation for COVID-19.

**Table 1 jcm-11-06025-t001:** Patient demographics 3–6 months after hospitalisation for COVID-19 stratified according to psychophysical olfactory test (Threshold, Discrimination, and Identification) results.

Variable	Normosmia (*n* = 21)	Hyposmia/Anosmia (*n* = 8)	*p*-Value
Gender, male (%)	67%	50%	*p* = 0.43
Age in years (SD)	53 (12.5)	62 (12) *p* = 0.13	*p* = 0.13
Admitted to hospital (due to COVID-19)	100%	100%	-
Admitted to the intensive care unit	24%	63%	*p* = 0.08
Duration of admittance in days (SD)	16.5 (16.0)	18 (13.0)	*p* = 0.77
Dyspnoea (%)	72%	75%	*p* = 1.00
FEV1 % predicted (SD)	111 (19.4)	109 (23)	*p* = 0.72
FEV1/FVC % (SD)	79 (5.1)	80 (7.7)	*p* = 0.80
Active smoker *	0%	0%	-
Former smokers (>6 months) *	57%	63%	*p* = 0.81
Remdesivir treatment **	5%	12%	*p* = 0.48
Dexamethasone treatment **	0%	0%	-
Chronic rhinosinusitis (CRS)	10%	13%	*p* = 1.00
Nasal secretion	48%	25%	*p* = 0.41
Nasal stenosis	10%	38%	*p* = 0.11
Sneezing	33%	38%	*p* = 1.00
Cough	33%	38%	*p* = 0.66
Postnasal secretion/drip	29%	38%	*p* = 0.66
Allergic rhinitis	38%	13%	*p* = 0.37
Nasal corticosteroids	10%	13%	*p* = 1.00
Per oral corticosteroids	5%	0%	*p* = 1.00
Normal nasal flexible endoscopy	53%	63%	*p* = 0.70
Non-obstructing nasal septum deviation	33%	25%	*p* = 1.00
Non-obstructing secretion/oedema/crusting	15%	0%	*p* = 0.54
Obstructing nasal septum deviation	5%	12%	*p* = 0.48
Concentration difficulties ~	19%	50%	*p* = 0.03
Headache	0%	0%	-
Charlson Comorbidity Score (CCS)			
0 (No risk of death within 1 year)	29%	0%	*p* = 0.15
1	10%	38%	*p* = 0.11
2	38%	12%	*p* = 0.37
3	5%	0%	*p* = 1.00
>3 (Highest risk of death within 1 year)	19%	50%	*p* = 0.16
Comorbidity	57%	75%	*p* = 0.67
Asthma	10%	0%	*p* = 1.00
Gastroesophageal reflux	5%	0%	*p* = 1.00

(SD): standard; *: N = 28; smoking status not available in 1 pt; **: during hospitalisation; ~: patients were asked if they experienced concentration difficulties. A *p*-value < 0.05 was considered statistically significant.

**Table 2 jcm-11-06025-t002:** Psychophysical olfactory and gustatory test results 3–6 months after hospitalisation for COVID-19.

Variable	All Patients (*n* = 29)
TDI score, mean (SD)	31 (8.3)
- Normosmia (TDI ≥ 30.75) (*n*), %	(22) 72%
- Hyposmia (TDI 17–30.74) (*n*), %	(6) 21%
- Anosmia (TDI ≤ 16) (*n*), %	(2) 7%
Threshold (T)score, mean (SD)	6 (2.3)
Discrimination (D) score, mean (SD)	11 (2.9)
Identification (I) score/SIT16, mean (SD)	13 (3.8)
- Normosmia (SIT16 > 11)%	79%
- Hyposmia (SIT16 9–11)%	11%
- Anosmia (SIT16 ≤ 8)%	11%
BSIT, mean (SD)	9 (2.8)
- Normosmia (BSIT > 9)%	64%
- Hyposmia (BSIT 8–6)%	18%
- Anosmia (BSIT ≤ 5)%	18%
Taste strips *	
- Normal (4/4)	88%
- Decreased (3/4)	12%
Taste strips incorrect answer	
- Salty %	3%
- Sour %	0%
- Sweet %	0%
- Bitter %	7%

* *n* = 25. TDI: Sniffin Sticks Threshold, Discrimination and Identification test. SD: standard deviation. SIT16: Sniffin Sticks Identification test 16. BSIT: Brief Smell Identification Test.

**Table 3 jcm-11-06025-t003:** Psychophysical olfactory test (Threshold, Discrimination, and Identification) results 12 months after hospitalisation for COVID-19 in patients with hyposmia/anosmia.

Variable	(*n* = 8)
Threshold (T) score, mean (SD)	4.3 (2.8)
Discrimination (D) score, mean (SD)	8.6 (3.0)
Identification (I/SIT16) score, mean (SD)	8.6 (3.7)
TDI score, mean (SD)	21.6 (6.9)
- Normosmia (TDI ≥ 30.75) (*n*), %	(0), 0%
- Hyposmia (TDI 17–30.74) (*n*), %	(6), 75%
- Anosmia (TDI ≤ 16) (*n*), %	(2), 25%

**Table 4 jcm-11-06025-t004:** a,b: Sensitivity and specificity for BSIT and SIT16 psychophysical identification tests compared with gold standard TDI.

**(a)**		**Threshold, Discrimination, Identification (TDI) Test**		
		**Hyposmia/** **Anosmia (+)**	**Normosmia (−)**	
Brief Smell Identification Test (BSIT)	Hyposmia/ anosmia (+)	7	3	Positive predictive value 7/10 × 100 = 70%
	Normosmia (−)	1	18	Negative predictive value 18/19 × 100 = 95%
		Sensitivity 7/8 × 100 = 87.5%	Specificity 18/21 × 100 = 85.7%	
**(b)**		**Threshold, Discrimination, Identification (TDI) Test**		
		**Hyposmia/** **Anosmia (+)**	**Normosmia (−)**	
Sniffin Sticks Identification Test (SIT16)	Hyposmia/ anosmia (+)	6	0	Positive predictive value 6/6 × 100 = 100%
	Normosmia (−)	2	21	Negative predictive value 21/23 × 100 = 91%
		Sensitivity 6/8 × 100 = 75%	Specificity 21/21 × 100 = 100%	

## Data Availability

Please contact the corresponding author.

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
