# Peer review of "Olfactory and Gustatory Outcomes Including Health-Related Quality of Life 3–6 and 12 Months after Severe-to-Critical COVID-19: A SECURe Prospective Cohort Study"

_jcm, 2022, doi:10.3390/jcm11206025_

Round 1

Reviewer 1 Report

The study presents findings on olfactory and gustatory outcomes after severe-to-critical COVID-19 in a small cohort of.

In my opinion, deep modifications are needed to make the study understandable and repeatable.

Firstly, methods are poorly described and it is not known which measurements were made in each study time (before/during-COVID-19, after 3-6 months, after 12 months). Besides, how did you make the choice of odors to be administered in the TDI? Did participants do the same testing during hospitalization as well? This description is important because a longitudinal study can be done if comparable measures are used over time: here the time pre-covid and during covid are investigated by an anamnestic questionnaire administered retrospectively (retrospective observational study and therefore subjective and not reliable, as you say at lines 237-240), while at follow-up after 3-6 months and after 12 months only TDI was done. 

Scores of the scales used are not explained, and statistical analysis is not provided. The Charlson comorbidity score (CCS) appears only in the Table 1 (?).

Data obtained from rhinoscopy not discussed, as well as possible selection bias related to sex (male prevalent). Variables related to comorbidities in Table 1 do not enter into the statistical analysis, so they might as well be removed.

A major revision of Tables is needed: tables should be self-explanatory without resorting to text. Captions are necessary, sometimes even layout (e.g., Table 4).

There are several typos in the text (e.g., does BSIT12 written in Results stand for BSIT written in Methods?, "incl." in the title; "olf." at line 154).

Author Response

Dear Reviewer

Reviewer 2 Report

The authors investigated olfactory and gustatory functions, and health-related quality of life in a group of previously hospitalized COVID-19 patients who reported olfactory and gustatory dysfunction 1 months after hospitalization. To this end, authors evaluated olfactory impairments using three different olfactory tests and compared their results.

This manuscript indicates that 28% of patients had OGD in a range of time between 3 and 6 months. I believe these results are of interest to the scientific community. However, I have some minor comments and questions listed below.

Material and Method

- Since it is well known that taste and olfaction can change during the day and in fasting condition (Stafford 2011, Chem Senses), what time of the day the tests were administered? Were the subjects requested to abstain from eating and drinking? Was the testing environment free from any odor? Please, specify in the section.

- Please specify the olfactory score for BSIT for normosmia, hyposmia and anosmia

- For a better comprehension, please specify that patients were tested in a single session between the 3rd and 6th month after discharge

- Please specify that the brief version of Taste Strips was used and what the gustatory score for normogeusia and hypogeusia were applied.

- Specify what essential oil were used for the olfactory training

- 101 Rehabilitation should be substituted with training for consistency

- Please describe how parosmia was assessed in patients

Results

- The percentage of patients with OD, GD, parosmia and dysgeusia are not consistent to the one expressed in the abstract. Please verify, and specify in the abstract which one of the olfactory tests were used for these classifications

- 131-34 and Fig 1: the reports of HRQoL are not clear, and what reported in the text is not consistent with the figure. Please explain better the results of the questionnaire

- Table 3: Authors should consider comparing the results of olfactory psychophysical tests 12 months after COVID-19 with the same tests performed from the 3rd to the 6th month after discharge

Author Response

Dear Reviewer

Reviewer 3 Report

Dear Authors,

In Your study You have analyzed olfactory and gustatory outcomes in previously hospitalized COVID-19 patients with different validated tests and questionnaire on QoL. I read Your work with interest. Here are my comments:

- I think the main drawback of Your study are the fact that patients were tested 3 to 6 months after discharge: this is a considerable span of time and as previous studies reported many patients undergo a spontaneous recovery of their smell and taste in the first 6 months. So it is possible that patients tested 6 months after discharge would have appeared anosmic if tested before. Another limitation is the fact that patients were not tested with psychophysical tests at baseline, so it is possible that some patients that declared themselves as "anosmic" and resulted normosmic at 3-6 months follow up were really never anosmic from the beginning. I think these critical aspects need to be highlighted in the paper;

- MATERIALS AND METHODS: You state that included patients had "severe-to-critical COVID-19". Please define what "severe-to-critical COVID-19" means;

- MATERIALS AND METHODS/RESULTS: patients were tested 3-6 months after discharge. Please consider to include more details on the average/median time when patients were tested in the results;

- MATERIALS AND METHODS/RESULTS: patients underwent flexible nasal endoscopy but there is no mention of that in the results except for Table 1;

- MATERIALS AND METHODS: the short sentence at line 100 "Gustatory dysfunction: <4/4." is not very clear to me;

- MATERIALS AND METHODS: at line 100 You anticipate and hyposmic/anosmic patients who were offered olfactory training were 8, however I think the number of patients belongs to the results should not be mentioned in the Materials and Methods section;

- MATERIALS AND METHODS: please include a paragraph on statistical analysis;

- RESULTS: I think it would be interesting to compare the normosmic to the hyposmic/anosmic group for all variables included in table 1 to see if there are significant difference between groups. (You could add a column in Table 1 to include CI and/or p value)

- RESULTS: I could not find the number of patients who subjectively reported themselves as hyposmic/anosmic at the first follow up in the results, please include that;

- RESULTS: regarding olfactory training, was there a statistically significant difference between TDI scores before and after the training? Please include that in Your paper;

- DISCUSSION: You state that hyposmic/anosmic patients had not improvement at the 12-month follow up. However this needs to be highlighted in the results with appropriate statistical tests to check Your hypothesis;

- DISCUSSION: lines 266-267: is this part of the results of Your study? There is no mention of that in the results nor in the methods section. Furthermore, statistical tests need to be applied in order to see if the difference is a statistically significant one. The same applies to lines 277-278;

- DISCUSSION: regarding patients with CRS (lines 283-288), I think this concept needs to appear also in the results of the study;

- DISCUSSION: concept at lines 293-295 needs to be clarifies in order to make is easily understandable by the reader;

- DISCUSSION: You mention that all patients underwent lung function testing, however there is no mention of that in the materials and methods section nor in the results (except for table 1), so please include that throughout the article.

Author Response

Dear Reviewer

Round 2

Reviewer 1 Report

Authors did a good revising work on my previous observations. However, some points should be revised, as follows:

1. There are some typos (e.g. "incl." at line 104, "precentages" at line 134, "Table 4a+b" at line 234).

2. Consider to write a "." [period] in place of the "," [comma] at line 125, just before "All tests were carried out [...]".

3. Methods are still poorly described in the issue of all variables inserted in Table 1: admission, dispnea, treatment, smoking status, nasal comorbiditis, headache and other comorbidity, ... are not mentioned in the Methods, while we read them only in Results. As previously said, there should be a correspondence between variables inserted in methods (in which they are identified and explained in their collection phase) and those in the Results. For example, these issues may be placed in a separate paragraph in the Methods section, in which you also explain why you chose them.

4. Methods: since the olfactory function score is based on scores from three different scales, specify how the classification in normosmia/hyposmia/anosmia was done. Did the scores concordants with each other? How did you consider any discrepancies between scores?

5. Methods: please include a proper citation of the statistical software (e.g., IBM Corp. Released 2017. IBM SPSS Statistics for Windows, Version 25.0. Armonk, NY: IBM Corp.).

6. Results: Please add the explanation of the black box in the caption of Figure 1.

7. Results: Table 2: SIT16 score is missing.

8. Table 3: Did you compare TDI mean scores between the first (3-6 months after COVID) and the second follow-up (12 months after COVID) in the overall sample and dividing by group (e.g., normosmia vs. hyposmia/anosmia, or olfactory training yes vs. no)? If yes, you may add results in Table 3.

9. Table 4: Authors should add notes explaining acronyms of TDI, SIT16, and BSIT (as previously said, table should be self-explanatory).

10. Results section should include also a paragraph explaining differences outlined in Table 1.

11. Discussion: the text at lines 316-349 should be revised as long as results from comorbidities need to be added in the Results section.

12. Please explain ENT at line 366.

Reviewer 3 Report

Dear Authors,

Thank You for reviewing Your Manuscript.

Author Response

Dear Reviewer 

We are glad that you were satisfied with our revision. 

Spelling mistakes have been corrected. 

kind regards